# Differences in Physical Demands and Player’s Individual Performance Between Winning and Losing Quarters on U-18 Basketball Players During Competition

**DOI:** 10.3390/jfmk9040211

**Published:** 2024-10-29

**Authors:** Adrià Miró, Jordi Vicens-Bordas, Marco Beato, Hugo Salazar, Jordi Coma, Carles Pintado, Franc García

**Affiliations:** 1Institut Nacional d’Educació Física de Catalunya (INEFC), Universitat de Barcelona, 08038 Barcelona, Spain; miroadria@gmail.com; 2Sport, Exercise and Human Movement (SEaHM) and Sport and Physical Activity Studies Centre (CEEAF), University of Vic-Central University of Catalonia, 08500 Barcelona, Spain; jordi.vicens@uvic.cat (J.V.-B.); jordi.coma@uvic.cat (J.C.); 3School of Allied Health Sciences, University of Suffolk, Ipswich IP4 1LQ, UK; 4Faculty of Education and Sport, University of Basque Country (UPV/EHU), 01007 Vitoria-Gasteiz, Spain; hsalazar002@gmail.com; 5Sports Performance Area, Futbol Club Barcelona, 08028 Barcelona, Spain; carles.pintado@fcbarcelona.cat; 6Barça Innovation Hub, Futbol Club Barcelona, 08028 Barcelona, Spain; 7Grup de Recerca en Ciències de l’Esport INEFC Barcelona (GRCEIB), Institut Nacional d’Educació Física de Catalunya (INEFC), Universitat de Barcelona, 08038 Barcelona, Spain

**Keywords:** physical demands, individual performance, quarter results, performance index rating, player load, steps, dynamic stress load

## Abstract

**Background:** This study examines how physical demands and individual performance influence quarter results in under-18 basketball players during a six-day basketball tournament. **Methods**: Twelve male players from an elite Spanish team were tracked using inertial microsensors to monitor external load variables (player load, steps, and dynamic stress load). Individual performance was assessed using the performance index rating (PIR). **Results**: The results showed significant differences in physical demands between quarters. Also, player load (F = 3.75, *p* = 0.012) and steps (F = 5.29, *p* = 0.001) were higher in the first quarter and decreased over time. Winning quarters had significantly higher physical demands compared to losing quarters (PL: F = 27.13, *p* < 0.001; steps: F = 16.70, *p* < 0.001; DSL: F = 9.50, *p* < 0.001). On the contrary, PIR did not show significant differences between winning and losing quarters (F = 2.15, *p* = 0.143), but tended to be higher in winning quarters. **Conclusions**: These results suggest that physical demands are stronger predictors of quarter results than individual performance scores, indicating that such parameters should be closely monitored by sport scientists and coaches since they can play a crucial role in team success.

## 1. Introduction

Basketball is a court-based team sport which is highly physically demanding, mostly due to repeated transitions between offense and defense and high-intensity movements, such as jumping, running, and shuffling [1]. Given the rigorous demands inherent in basketball gameplay, it becomes paramount for practitioners to meticulously track the physical stressors experienced by players. In the pursuit of performance optimization, careful attention must be given to the physical demands (e.g., external load parameters) as a potential driver for increasing player availability and enhancing in-game performance [2].

Physical and performance demands refer to the physical, technical, and tactical activities undertaken during training and competition, determined by factors such as the organization, quality, and quantity of exercises [3]. Physical demands can be assessed using some external load parameters; specifically, they can be measured through tracking technologies such as radar-based local positioning systems (LPS), global positioning systems, and multiple-camera video technology [4]. These technologies enable the recording of metrics such as distance covered, player load (PL), steps, accelerations, dynamic stress load (DSL), and activity duration [3]. External load parameters offer complementary insights into players’ physical load and performance during games, enabling coaches and sports scientists to better understand training needs and optimize physical and tactical preparation strategies [1,5].

In basketball, the physical demands vary significantly by player position. Point guards tend to cover more distance, execute more accelerations, and perform rapid changes of direction due to their role in facilitating transitions, requiring high agility and endurance [1]. Conversely, centers engage more in physical contacts, rebounds, and short, intense sprints, emphasizing strength and power [6,7]. Comparatively, sports like handball and futsal, with similar high-intensity demands and small spaces, offer valuable insights into understanding these positional differences and their impact on external load metrics [8,9].

During basketball competitions, the physical demands on players may vary between the first and fourth quarters of a game, but it is not yet clear whether the physical demands decrease at the end of basketball games [10]. While some studies have examined distance and speed parameters, they did not find any clear physical decrement in the final quarter [11,12]; however, other studies reported a significant decline in high-intensity actions (e.g., accelerations, decelerations, distances covered at high speed and at maximal running speed) between quarters [6,7,13] and player workload [6]. Therefore, more research is needed to understand the fatigue phenomenon in basketball and how fatigue affects individual and team performance towards the end of games (i.e., differences between quarters).

For sports scientists and coaches, understanding the associations between external load parameters and performance outcomes is critical to optimizing training programs to increase the likelihood of success for their team [14]. For instance, typical basketball performance parameters are successful shooting percentage, number of assists, and mastery of defensive rebounding, which are key performance indicators for team success [15]. Referring to individual player performance, one of the most common metrics in European basketball is the performance index rating (PIR), which combines all the positive and negative aspects of each player for an individual performance assessment score [16].

Previous research investigated the relationships between physical demands, weekly training, game load, and basketball performance measured through in-game statistics [17,18,19]. It was demonstrated that the peak physical demands during a quarter in basketball had little to no effect on the performance displayed by players on the court [20]. However, no research has examined the influence of physical demands between winning and losing quarters and its relationship with basketball game performance so far. Therefore, the aim of this study was to deepen the understanding of how physical demands influence the individual performance of U-18 male basketball players during game quarters. The authors of this research hypothesized that the players’ physical demands would not affect their individual performance across game quarters. However, they expected that higher external load parameters would be associated with quarters won by the team compared to losing quarters.

## 2. Materials and Methods

### 2.1. Participants

Twelve male basketball players (age: 16.8 ± 1.7 years; height: 198 ± 11 cm; body mass: 80.4 ± 7.6 kg; experience: 8.5 ± 1.7 years) participated in this study. The participants were reserve players of a Spanish Euroleague team who competed simultaneously in the under-eighteen (U-18) Catalan regional league and in the Spanish fourth division league (Liga EBA) from September 2021 to May 2022. All the participants remained free of injuries during the data collection period.

The study was conducted in accordance with the principles of the Helsinki Statement of 2008, updated in Fortaleza, October 2013 [21]. Before the study began, all the players received information about the study and gave their signed agreement to participate. Players were routinely observed throughout the whole competition season; therefore, following Winter and Maughan’s guidelines, an ethics committee clearance was not necessary for this study [22].

### 2.2. Design

A retrospective observational study was conducted to quantify the physical demands, and player’s individual performance encountered according to game quarter (first vs. second vs. third vs. fourth), and quarter result (win vs. tie vs. loss). All games were part of the U-18 Spanish National Championship played in Huelva (Spain) in the afternoon between 1 May and 7 May 2021. Games were played following the official International Basketball Association rules.

For inclusion in the study, players had to actively participate in each quarter for at least one minute, and there was no exclusion criteria applied.

### 2.3. Methodology

Six different game samples were gathered throughout the competition (i.e., each player was analyzed on multiple occasions). Throughout the course of six days, participants were involved in one game per day. Before every game, players engaged in a standardized 45 min warm-up that included muscular activation, dynamic stretching, mobility drills, and basketball particular actions, such shooting, passing, and dribbling.

Players’ physical demands were continuously monitored with a local positioning system (WIMU PRO™, RealTrack Systems SL, Almería, Spain). During the games, the system (with the following physical characteristics: 81 × 45 × 15 mm, and 70 g.) was placed in the center of the player’s upper back using an adjustable custom-made harness (IMAX, Mississauga, ON, Canada). To limit intersystem variability [23], each participant wore the same assigned device in all six games. The system-specific software, SPRO™ (version 987, RealTrack Systems, Almería, Spain), was used to download and analyze data on the players’ physical demands and physiological responses.

The physical variables measured were PL, expressed in arbitrary units (AU) and calculated as the sum of the squared rates of change in acceleration (also known as jerk) in each of the three vectors divided by 100 [24]; number of total steps; and DSL, which is an accelerometer-derived metric that quantifies the cumulative stress placed on an athlete’s body during sports activities [25]. DSL is calculated by aggregating the acceleration rates along three orthogonal axes (X, Y, Z), forming a composite vector expressed in G-force [26]. DSL complements PL by providing a more detailed analysis of the specific mechanical stress imposed during rapid directional changes and explosive movements [27]; furthermore, to our knowledge, DSL has been minimally explored in basketball research [19]. Additionally, the inclusion of steps offers valuable insight into the overall volume of activity, allowing for a comprehensive assessment of both movement intensity and quantity. These metrics’ absolute values were standardized by dividing them by each player’s total playing time (in minutes). Breaks between quarters, timeouts, and substitution time were excluded to avoid collecting physical data such as steps or PL unrelated to the game demands [11]. Playing time was defined as the time during which players were competing on the court, including in-game stoppages, such as between fouls and free throws.

Players’ individual performance was determined using PIR [17,28], which was collected by an official independent score technician working for the Spanish Basketball Federation. PIR scores were also normalized by the playing time spent on the court (e.g., a PIR of 10 obtained during 20 min of total playing time was 0.5). PIR was calculated using the following formula:PIR = (points + rebounds + assists + steals + blocks + fouls drawn) − (missed field goals + missed free throws + turnovers + shots rejected + personal fouls made). (1)

### 2.4. Statistical Analysis

The data were presented as estimated marginal means with 95% confidence intervals (95% CIs) for each dependent variable. The normality assumption of the residuals was assessed using the Kolmogorov–Smirnov test and was met for the parameters analyzed. Therefore, the dependent variables were analyzed using linear mixed models to account for missing data and repeated measures. Separate models for each dependent variable were used with the game day (i.e., from 1 to 6) entered as fixed effects, while the player was entered as a random effect into each model. Statistical significance was set at *p* < 0.05. Pairwise comparisons with the Bonferroni correction were performed using post hoc tests. The *t* statistics from the mixed model were converted into Cohen’s *d* effect sizes (ES), with associated 95% CIs. The magnitude of differences in each pairwise comparison was assessed using ES analysis with 95% confidence intervals. ESs were interpreted as: <0.20 = trivial; 0.20–0.59 = small; 0.60–1.19 = moderate; 1.20–1.99 = large; and >2.00 = very large [29]. Contrasts, which are techniques that compare groups or treatments using linear combinations of parameters to assess whether there are significant differences among them, are displayed in the Appendix A. Analysis was conducted using the jamovi package (The jamovi project (2023). jamovi (Version 2.3.26) [Computer Software]. Retrieved from https://www.jamovi.org (accessed on 10 September 2023)).

## 3. Results

The team finished the competition with a five wins–one lose record, losing the penultimate game. No injuries were suffered by the players throughout any of the matches. The results are summarized in Table 1 and Table 2, and Appendix A.

Differences between game quarters were found for PL (F = 3.75, *p* = 0.012) and steps. (F = 5.29, *p* = 0.001). Differences were observed among the first and the fourth quarter for PL (*0.11 au·min*^−1^, *d* = 0.42) and steps (7.8 steps*·min*^−1^, *d* = 0.52) (see Appendix A). Also, differences were present between the first and the second quarter for PL (*0.11 au·min*^−1^, *d* = 0.37) and steps (7.8 steps*·min*^−1^, *d* = 0.41). No significant differences were found between game quarters for DSL (F = 1.50, *p* = 0.216) and PIR (F = 0.24, *p* = 0.864).

Concerning winning or losing the quarter, differences were found for all the external load metrics: PL (F = 27.13, *p* ≤ 0.001); steps (F = 16.70, *p* ≤ 0.001); and DSL (F = 9.50, *p* ≤ 0.001). Also, the analysis revealed that winning the quarter resulted in higher physical demand outcomes compared to losing (see Appendix A): PL (*0.14 au·min*^−1^, *d* = 0.72); steps (8.03 steps*·min*^−1^, *d* = 0.70); and DSL (*0.53 au·min*^−1^, d = 0.47). In addition, contrasts indicated differences between winning the first and the fourth quarter for steps (9.7 steps*·min*^−1^, *d* = 0.49) and for winning and losing the second quarter (11.77 steps*·min*^−1^, *d* = 0.58). Moreover, differences were present between winning and losing the second quarter for PL (*0.22 au·min*^−1^, *d* = 0.65).

No significant differences were observed between winning or losing a quarter for PIR (F = 2.15, *p* = 0.143); however, the PIR was higher in quarters that were won (mean = 2.60) compared to those that were lost (mean = 1.88).

Lastly, no significant differences were found between PIR and PL (F = 0.007, *p* = 0.933), PIR and steps (F = 0.46, *p* ≤ 0.499), or PIR and DSL (F = 1.36, *p* = 0.248).

## 4. Discussion

The aim of this study was to investigate the influence of external load parameters on players’ individual performance in U-18 male basketball players during game quarters. In addition to players’ PIR, the present research examined the fluctuations of PL, steps, and DSL during a six-day official basketball competition among elite U-18 basketball players. Such findings are essential to facilitating the adjustment of training recommendations to enhance a player’s individual and team performance. The primary finding of this study was that there was a clear downward trend throughout game quarters for all the external load parameters analyzed and differences between winning or losing the quarter. Specifically, when a team won a quarter, the external load was higher compared to when the team lost the quarter. In addition, no significant differences in PIR were observed between winning and losing quarters.

The current study recognized significant differences among game quarters regarding PL and steps normalized by the total playing time. Similar to the existing literature [6,30], the greatest differences between game quarters were found between the first and the fourth quarter. This reduction in players’ external load parameters may suggest physiological fatigue as the game progresses, potentially leading to a decline in their physical performance [31]. In addition to fatigue, other factors like tactical strategies may come into play, with teams slowing the game pace in later quarters to maintain better ball control and increase the chances of a successful outcome [13]. Also, the present research indicated significant differences between the first and the second game quarter [30] for PL and steps. On the contrary, no statistically significant differences were observed between game quarters for DSL and PIR. This study shows that PL significantly changes during the match while DSL does not, which suggests that the PL and DSL assess different mechanical load aspects [27]. Both PL and DSL have been previously validated (i.e., convergent validity) by correlation with measures of internal physiological load, exercise intensity, total distance, collisions and impacts, fatigue, and injury risk and incidence. Thus, they have passed a process of validation that allows their use in sport performance monitoring [27]. Specifically, Beato and colleagues showed DSL utility in monitoring intermittent sports activities (e.g., repeated sprint and shuttle runs), which highlights its possible use for the monitoring of sports that use the same intermittent running pattern [32]. Regarding the use of DSL in basketball [19], many more teams are using this external load parameter nowadays (and other mechanical-related parameters monitored by wearable technology) [27]; however, the scientific evidence for their use is very limited and much more research is needed.

In addition to the relationship between external load and game quarters, our results on PIR and physical demands agree with the existing literature, where research has shown that the peak physical demands during a quarter in basketball has little to no effect on the individual players’ performance on the court [17,28]. With respect to winning or losing quarters, external load metrics (i.e., PL, steps, and DSL) were higher in quarters that were won compared to those that were lost. When comparing our results to the current basketball literature, we observed resemblances with the study conducted by [20], who observed that quarters in which the team won or tied exhibited slightly higher physical demands, when compared to quarters that ended in a loss. On the contrary, no significant differences in PIR were observed between winning and losing quarters (*p* = 0.143). However, it is noteworthy that PIR was higher when the quarter were won (mean = 2.60) rather than when they were lost (mean = 1.88). This suggests a possible correlation between winning and higher individual player performance (measure using PIR). However, further research is needed, as no previous studies have examined the relationship between winning a quarter and a higher PIR.

This study is not without limitations. Firstly, the participants were under 18, so it may not be generalizable to other forms of basketball competition or senior teams and players. Secondly, the sample size was small (*n* = 12), and the tournament spanned 6 days of games. Thirdly, the absence of a female sample restricts the generalizability of our findings, and distinguishing player roles could provide further insights into the data. Furthermore, no internal load (or physiological demands) parameters were measured to complement the external load parameters, limiting a more comprehensive understanding of players’ match performance and fatigue. Nevertheless, this study was performed during an official tournament which represents an ecological scenario where physiological parameters are not usually monitored to avoid interference with the players’ performance. Despite these limitations, our findings provide valuable insights into the external load parameters and their impact on the performance of U-18 basketball players during game quarters. Future research should aim to include larger sample sizes and a more extended period of observation, and a more diverse participant pool while analyzing the impact of different player roles to validate these findings and explore additional metrics that may influence performance outcomes.

## 5. Conclusions

This study reveals important distinctions in the physical demands across different quarters and their impact on game outcomes in U-18 basketball. The external load parameters, such as PL and steps, were significantly higher in earlier quarters and in quarters that were won, suggesting that physical demands are a key factor in winning. Moreover, no significant differences were found in the PIR across game quarters or between winning and losing quarters, although a non-significant but higher PIR was observed in quarters that were won (mean = 2.60) compared to those that were lost (mean = 1.88). Future studies could explore a potential correlation between winning quarters and increased individual player performance. This study suggests the need for a more systematic approach to understanding basketball performance, which would require sports scientists and coaches to incorporate additional factors beyond external load parameters to better assess and optimize player performance and game outcomes.

## Figures and Tables

**Table 1 jfmk-09-00211-t001:** Estimates of interactions between game quarter (number and win/lost) and PIR.

Game Quarter	Win/Lost	Mean (PIR)	SE	95% CI Lower	95% CI Upper
1Q	Win	2.72	0.55	1.6	3.85
	Lost	-	-	-	-
2Q	Win	2.41	0.61	1.18	3.65
	Lost	2.04	0.86	0.32	3.76
3Q	Win	2.45	0.60	1.25	3.65
	Lost	1.13	1.12	−1.08	3.35
4Q	Win	3.01	0.86	1.30	4.73
	Lost	2.02	0.68	0.66	3.38

**Table 2 jfmk-09-00211-t002:** Estimates of interactions between game quarter (number and win/lost) and physical demand metrics.

Game Quarter	Win/Lost	PL $\bar{X}$ (SE) [95% CI]	Steps $\bar{X}$ (SE) [95% CI]	DSL $\bar{X}$ (SE) [95% CI]
1Q	Win	1.35 ± 0.03	74.51 ± 2.09	3.31 ± 0.50
		[1.26–1.43]	[70.17–78.86]	[2.09–4.31]
	Lost	-	-	-
		-	-	-
2Q	Win	1.32 ± 0.04	72.14 ± 2.24	3.22 ± 0.51
		[1.23–1.41]	[67.56–76.71]	[2.11–4.33]
	Lost	1.1 ± 0.05	60.37 ± 2.87	2.20 ± 0.54
		[0.99–1.20]	[54.64–66.11]	[1.05–3.35]
3Q	Win	1.31 ± 0.04	71.93 ± 2.19	3.20 ± 0.54
		[1.22–1.39]	[67.44–76.42]	[2.44–4.75]
	Lost	1.14 ± 0.06	58.88 ± 3.56	3.20 ± 0.58
		[1.02–1.27]	[51.83–65.92]	[1.99–4.41]
4Q	Win	1.24 ± 0.05	64.87 ± 2.86	3.60 ± 0.54
		[1.14–1.34]	[59.17–70.58]	[2.44–4.75]
	Lost	1.24 ± 0.04	67.71 ± 2.39	2.94 ± 0.52
		[1.15–1.33]	[62.88–72.55]	[1.82–4.05]

## Data Availability

This study was carried out with the data obtained from the FINA page, so informed consent was not needed, since they are public data.

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
