# Peer review of "Differences in Physical Demands and Player’s Individual Performance Between Winning and Losing Quarters on U-18 Basketball Players During Competition"

_jfmk, 2024, doi:10.3390/jfmk9040211_

Round 1
Reviewer 1 Report
Comments and Suggestions for Authors
The manuscript offers an insightful analysis of the relationship between physical demands and individual performance in U-18 basketball players during competition. The study is thoughtfully designed with clear objectives, and it provides useful information for sports scientists and coaches on how physical demands may influence the outcome of game quarters. That said, there are a few minor issues that need to be addressed:
1. Line 17: I think you can’t say »affect« here. Probably performance and physical demands are affecting the outcome (win/lose), not the other way around.
2. Results: I suggest you used “d” instead of “ES” in results text
3. Table 1 and 2 are difficult to understand. What does the mean value represent here?
4. What was the total numerus of the player-quarter units? If all players played all quarters for at least one minute, it is 4 quarters x 6 games x 12 players = 288. Is that the case?
5. This is just a suggestion, but instead of win/lose categorization, why not have the score difference as one variable (e.g., a lose could be for -1 point or -10). Then you could also run a correlation analysis for further insights.
Author Response
The manuscript offers an insightful analysis of the relationship between physical demands and individual performance in U-18 basketball players during competition. The study is thoughtfully designed with clear objectives, and it provides useful information for sports scientists and coaches on how physical demands may influence the outcome of game quarters.
Authors’ response: We thank the reviewer for the kind comment.
1. Line 17: I think you can’t say »affect« here. Probably performance and physical demands are affecting the outcome (win/lose), not the other way around.
Authors’ response: Thank you for your feedback. You're absolutely right. Based on your suggestion, we have reworded the sentence to reflect that physical demands and individual performance influence the quarter results. The revised sentence now reads: "This study examines how physical demands and individual performance influence quarter results in under-18 basketball players during a six-day basketball tournament.".
2. Results: I suggest you used “d” instead of “ES” in results text.
Authors’ response: Thank you for your suggestion. We have made the change, replacing "ES" with "d" in the results section.
3. Table 1 and 2 are difficult to understand. What does the mean value represent here?
Authors’ response: Thank you for your comment. The "mean" value in the tables represents the interaction between the quarter (whether it was won or lost) and the variable being studied. In Table 1, this variable is the Performance Index Rating (PIR), while in Table 2, it refers to the physical demands studied in the article, specifically PlayerLoad (PL), steps, and Dynamic Stress Load (DSL). We have made a small adjustment to the table 1 that I believe will help improve its clarity. Please feel free to ask if you have any further questions.
4. What was the total numerus of the player-quarter units? If all players played all quarters for at least one minute, it is 4 quarters x 6 games x 12 players = 288. Is that the case?
Authors’ response: The total number of player-quarter units is 229, following the inclusion criteria. Not all players played in every quarter for at least one minute, which explains the difference from the 288 units.
5. This is just a suggestion, but instead of win/lose categorization, why not have the score difference as one variable (e.g., a lose could be for -1 point or -10). Then you could also run a correlation analysis for further insights.
Authors’ response: Thank you for your suggestion. It’s actually an idea we had in mind when designing the study. Initially, we wanted to categorize wins and losses based on whether the quarter or the overall game had a close or wide score difference. However, when it came to the analysis and presentation of the results, this approach became more complex and harder to interpret. For this reason, we decided to simplify the categorization to just win or lose.

Reviewer 2 Report
Comments and Suggestions for Authors
This article deals with a very interesting topic with high predictability of applicability. Some comments have been added below:
Introduction
More details should be provided on this physiological requirement regarding the role of each player (e.g., the distinction of the point guard with respect to the center).
In this part of the paper, to contextualize and cover the lack of specific literature, we could add works on sports with similar demands and spaces (handball, indoor soccer, etc.).
Methods
The code of the ethics committee is not included in this paper. The experience of the sample in years of playing basketball should be specified.
Perhaps it could be added to the limitation that there was no female sample with which to compare these results.
Discussion
Although it is logical that in the winning quarter of a game, there is greater demand, what happens if the team is technically superior? This premise of a higher demand is probably not met. Perhaps it should indicate the level of the team according to the ranking; this variable can help better interpret the results.
Limitations include the lack of a female sample and the lack of distinguishing player roles.
Author Response
This article deals with a very interesting topic with high predictability of applicability. Some comments have been added below.
Authors’ response: Thank you for your positive feedback! We are glad you find the topic interesting and applicable. We appreciate your detailed comments and suggestions.
Introduction
More details should be provided on this physiological requirement regarding the role of each player (e.g., the distinction of the point guard with respect to the center). In this part of the paper, to contextualize and cover the lack of specific literature, we could add works on sports with similar demands and spaces (handball, indoor soccer, etc.).
Authors’ response: Thank you for your insightful comment. We have expanded the section to provide more details on the specific physiological demands associated with different player roles, such as the distinction between the point guard and the center, as their tactical and technical responsibilities result in varying physical demands. Additionally, we have incorporated references to sports with similar demands and spaces, such as handball and futsal, to better contextualize our findings in the absence of specific literature.
Methods
(1) The code of the ethics committee is not included in this paper.
(2) The experience of the sample in years of playing basketball should be specified.
(3) Perhaps it could be added to the limitation that there was no female sample with which to compare these results.
Authors’ response:
(1) Since the players were routinely observed throughout the entire competition season as part of their regular activities, ethics committee clearance was not necessary for this study. We have reported the following indication in the text: The study was conducted in accordance with the principles of the Helsinki Statement of 2008, updated in Fortaleza, October 2013 [24]. Before the study began, all players received information about the study and gave their signed agreement to participate. Players were routinely observed throughout the whole competition season; therefore, ethics committee clearance was not necessary [25].
(2) Thank you for the suggestion. We have now added the information regarding the experience of the sample to the original text. The average experience of the participants in years of playing basketball is 8.5 ± 1.7.
(3) Thank you for highlighting these important limitations. We acknowledge that the absence of a female sample restricts the generalizability of our findings and that distinguishing player roles could provide further insights into the data. Future research will aim to address these gaps by including a more diverse participant pool and analyzing the impact of different player roles on the results.
Discussion
(1) Although it is logical that in the winning quarter of a game, there is greater demand, what happens if the team is technically superior? This premise of a higher demand is probably not met. Perhaps it should indicate the level of the team according to the ranking; this variable can help better interpret the results.
(2) Limitations include the lack of a female sample and the lack of distinguishing player roles.
Authors’ response:
(1) Thank you for your insightful comment. We agree that the result of the quarter is likely influenced by multiple contextual factors. The technical superiority of the team could potentially mean that players require less physical effort to achieve their goals, or it could be that this factor does not influence the physical demand at all. We are not certain.
Regarding the level of the team according to ranking, it is difficult to provide specific details since there is no official ranking in the Spanish U18 championship. However, we do know that the team analyzed was expected to win all the games they did win, and they lost only against a team with a similar level of ability. We hope this context helps clarify the interpretation of the results.
(2) Thank you for your comment. We have already addressed these limitations in the revised manuscript. The discussion now acknowledges the absence of a female sample and highlights the need to distinguish player roles for a more comprehensive understanding of the data.
